# Cell Stress Induces Mislocalization of Transcription Factors with Mitochondrial Enrichment

**DOI:** 10.3390/ijms22168853

**Published:** 2021-08-17

**Authors:** Chiara Rossi, Anna Fernàndez, Pascual Torres, Omar Ramirez-Nuñez, Ana Belén Granado-Serrano, Laia Fontdevila, Mònica Povedano, Reinald Pamplona, Isidro Ferrer, Manuel Portero-Otin

**Affiliations:** 1Metabolic Physiopathology Research Group, Experimental Medicine Department, Lleida University-Lleida Biochemical Research Institute (UdL-IRBLleida), E25198 Lleida, Spain; clara70289@mex.udl.cat (C.R.); anna.fernandez@udl.cat (A.F.); pascual.torres@udl.cat (P.T.); omar.ramirez78@gmail.com (O.R.-N.); anabgs@mex.udl.cat (A.B.G.-S.); laia.fontdevila@mex.udl.cat (L.F.); reinald.pamplona@udl.cat (R.P.); 2Functional Unit of Amyotrophic Lateral Sclerosis (UFELA), Service of Neurology, Bellvitge University Hospital, L’Hospitalet de Llobregat, E08907 Barcelona, Spain; 30058mpp@gmail.com; 3Department of Pathology and Experimental Therapeutics, University of Barcelona, E08900 Barcelona, Spain; 8082ifa@gmail.com; 4CIBERNED (Network Centre of Biomedical Research of Neurodegenerative Diseases), Institute of Health Carlos III, Ministry of Economy and Competitiveness, E08900 Barcelona, Spain; 5Bellvitge Biomedical Research Institute (IDIBELL), Hospitalet de Llobregat, E08907 Barcelona, Spain; 6Senior Consultant, Bellvitge University Hospital, E08900 Barcelona, Spain; 7Institute of Neurosciences, University of Barcelona, E08000 Barcelona, Spain; 8Edifici Biomedicina I, Avda Rovira Roure, E25196 Lleida, Spain

**Keywords:** TDP-43, Jun, REST, ERK, mitochondria, cell stress, aggregation, transcription factors, transgenic mice, subcellular fractionation

## Abstract

Previous evidence links the formation of extranuclear inclusions of transcription factors, such as ERK, Jun, TDP-43, and REST, with oxidative, endoplasmic-reticulum, proteasomal, and osmotic stress. To further characterize its extranuclear location, we performed a high-content screening based on confocal microscopy and automatized image analyses of an epithelial cell culture treated with hydrogen peroxide, thapsigargin, epoxomicin, or sorbitol at different concentrations and times to recreate the stresses mentioned above. We also performed a subcellular fractionation of the brain from transgenic mice overexpressing the Q331K-mutated TARDBP, and we analyzed the REST-regulated mRNAs. The results show that these nuclear proteins exhibit a mitochondrial location, together with significant nuclear/extranuclear ratio changes, in a protein and stress-specific manner. The presence of these proteins in enriched mitochondrial fractions in vivo confirmed the results of the image analyses. TDP-43 aggregation was associated with alterations in the mRNA levels of the REST target genes involved in calcium homeostasis, apoptosis, and metabolism. In conclusion, cell stress increased the mitochondrial translocation of nuclear proteins, increasing the chance of proteostasis alterations. Furthermore, TDP-43 aggregation impacts REST target genes, disclosing an exciting interaction between these two transcription factors in neurodegenerative processes.

## 1. Introduction

The presence of protein aggregates is a pathological hallmark for several neurodegenerative diseases, including Alzheimer’s disease (AD), Parkinson’s disease, and amyotrophic lateral sclerosis (ALS), to name a few [1]. These protein aggregates show disease specificity. For instance, beta-amyloid aggregates characterize the AD neurofibrillary tangles and intracellular aggregates. Similarly, aggregates of highly phosphorylated TDP-43 are present in the cytosol of the remaining motor neurons in ALS. Whether (and how) these aggregates are mere bystanders of ongoing pathogenic processes or whether they constitute bona fide cellular noxa is still under debate.

In the case of ALS, TDP-43 is a primary component of these aggregates. TDP-43 is a ubiquitous protein that belongs to the heterogeneous nuclear ribonucleoprotein family and is encoded by the TARDBP gene. TDP-43 is mainly found in the nucleus of normal cells and is involved in RNA regulation, including transcriptional regulation, alternative splicing, and mRNA stabilization [2]. TDP-43 is usually found in the nucleus, but it also moves between the nucleus and the cytoplasm to perform various cellular tasks. TDP-43’s level and localization are tightly controlled by a negative feedback mechanism [3]. Under stress circumstances such as heat shock, oxidative stress, and arsenite exposure, nuclear TDP-43 is transferred to the cytoplasm, and cytoplasmic TDP-43 accumulates to form stress granules (SGs), a variety of other proteins and RNAs [4]. When the stress is relieved, the SGs that carry TDP-43 break down, and TDP-43 released from the SGs translocates into the nucleus [5]. On the other hand, chronic stress causes extended SG formation, which leads to cytoplasmic TDP-43 aggregate accumulation.

In addition to TDP-43, other proteins with roles as transcriptional factors also show their cytoplasmic accumulation as aggregates [6]. We included p-ERK [7] and p-Jun [8]. Protein aggregation has been found in several neurodegenerative conditions, such as AD. Interestingly, recent data has demonstrated that another protein, REST, could show decreased values during AD pathogenesis [9]. Similarly, REST is found as aggregates in the substantia nigra in PD patients [10]. In this context, it is known that cellular stress, such as ER, proteasome, or oxidative stress, can induce TDP-43 mislocalization [7].

The dysfunction of the nuclear pore complex is linked to cytoplasmic mislocalization and TDP-43 aggregation, in addition to the dysregulation of stress granule formation [11]. Indeed, one of the most common causes of ALS, associated with G4C2 repeats within the C9ORF72 gene, impairs the cytoplasm–nucleus gradient of Ran, the primary regulator of TDP-43 nuclear localization, according to several studies [12]. The structure of the nuclear membrane is similarly disrupted by (G4C2) RNA. TDP-43 is also linked to the cytoplasmic accumulation of other nuclear membrane proteins, like Nup62 and Kpnb1 [13]. Of note, we have recently demonstrated that ALS is associated with an alteration in the nuclear envelope lipids [14].

Additionally, p-TDP-43, p-ERK [7], and REST [10] also show aggregates in neurodegenerative diseases. Several authors have reported that TDP-43 protein aggregates may be found in organellar fractions, in addition to the cytosolic location, such as mitochondria [15,16]. How cell stress is linked to this mislocalization is currently unknown, but it may include impairments in proteostasis and imbalances in autophagy flux [17]. To shed light on this question, we have explored if oxidative, ER, and proteasome stress can induce changes in the cellular distribution of these transcription factors. Specifically, we focused on the potential interaction with mitochondria due to the relevance of mitochondrial (dys)function in neurodegenerative conditions (particularly in ALS). We validated the in vitro data by exploring the presence of these proteins in enriched mitochondrial fractions. Our results demonstrate that the interactions of TDP-43, REST, Jun, and ERK with the extranuclear components, such as mitochondria, are dependent on cell stress. Further, we demonstrate an unreported association between TDP-43 pathology and the changes in REST-dependent mRNAs. All in all, the enhanced mitochondrial interaction of these proteins could contribute to the reported loss of mitochondrial functions in several neurodegenerative processes.

## 2. Results

Oxidative stress induced by H_2_O_2_ in HMEC cells led to changes in the nuclear and non-nuclear distribution of p-TDP-43, while the nuclear intensity of p-TDP-43 was decreased in the milder oxidative conditions tested (Figure 1A and Appendix A). In these milder conditions, the cell viability was preserved (Appendix A). In contrast with p-TDP-43, despite an initial increase in a p-ERK cytosolic area in the same conditions, this protein was rapidly cleared (Figure 1A), being mainly non-nuclear. The cytoplasmic staining of p-Jun and REST showed the same tendencies, i.e., after an initial decrease, there was a tendency for increasing their values (highly significant in the case of REST), as indicated in Figure 1A. The nonhomogeneous distribution of cytoplasmic locations after oxidative stress (evident in p-TDP-43, p-Jun, and REST) suggested their colocalization with an organelle fraction. We performed coimmunostaining with several mitochondrial epitopes to test if this non-nuclear localization involved a mitochondrial residence (as previously indicated in reference [18]). Complex V coimmunostaining with these factors (Figure 1B) indicated that the degree of colocalization increased significantly in all the cases after oxidative stress, exceeding z’-values of 0.5 (roughly meaning that at least 50% of both epitopes could coincide at the resolution of the confocal microscopy), except for p-ERK. Although the colocalization of this protein increased significantly after oxidative stress (Figure 1B), and in line with the p-ERK decreased values in the cytoplasm, the z’-values did not reach 0.3 in this case. The mitochondrial networks were significantly affected by oxidative stress. Thus, while low concentrations of hydrogen peroxide did not impinge changes in the mitochondrial number or in its networking, higher concentrations, associated with the changes in TDP-43 and other transcription factors, were associated with the increased mitochondrial numbers and networks (Appendix A). The correlation analyses of nuclear vs. cytoplasmic intensity showed linear relationships between these parameters, though their responses to oxidative stress were strongly dependent on the factor (Appendix A).

These immunostaining results suggested a partial colocalization of the mitochondrial epitopes and ALS-related protein factors in a cell line. To validate these results in an independent setup, we evaluated this phenomenon in a murine model of ALS-related neurodegeneration. We employed TDP-43 Q331K mice overexpressing the human mutated TDP-43 gene. The results of the subcellular fraction in the brain agree with the in vitro findings. Thus, both TDP-43 and ERK are present in crude mitochondria, a subcellular fraction enriched in mitochondria but also containing other membranes (Figure 2 and Appendix A). Indeed, as expected, h-TDP-43 was highly enriched in the cytosol, nuclei, and crude mitochondria fraction. Demonstrating that TDP-43 location in the crude mitochondria was not an artifact of the overexpression of this gene, endogenous (murine) Tdp-43 was also present, as evidenced by Western blotting (Figure 2). The analyses of variance demonstrated that the TDP-43 amount was significantly affected by transgenesis (22% of the total variation, *p* < 0.001, after a three-way ANOVA, Appendix A). In contrast, the subcellular location strongly influenced the p-TDP-43 levels (47% of the variation, *p* < 0.0001), followed by sex (13.6% of the variation, *p* < 0.0035), but not the transgenesis (Figure 2 and Appendix A). In general, the values of p-TDP-43 were higher in female mice (Figure 2 and Appendix A for males), and in both genders, the levels of p-TDP-43 were higher in the crude mitochondria than in cytosol (Figure 2).

Concerning ERK, the findings in the spinal cord of that murine model reinforce the in vitro data. Thus, in line with the colocalization of p-ERK with complex V, the crude mitochondria showed a relevant concentration of p-ERK. However, the nuclear fractions showed a high concentration of p-ERK, while the total ERK levels were significantly lower in the nuclei than cytosol (Figure 2). Densitometry analyses indicated that the total ERK levels were not significantly influenced by the sex, transgene expression, or subcellular fraction (Appendix A). However, when evaluating p-ERK, the subcellular location influenced, to the greatest extent, its levels (52% of the variation, *p* < 0.0001), with sex interacting with the subcellular location also being a relevant factor (13.8%, *p* < 0.008).

Regarding p-Jun, we detected its presence mainly in nuclear fractions, with almost no detection in the crude mitochondrial fraction or cytosol. After densitometry, most of the variance was explained by the subcellular location (63% of the variance, *p* < 0.0001), with sex also influencing the values (9.6%, *p* < 0.006). TDP-43 overexpression only explained 4% of the total variance in the sex and subcellular location (*p* < 0.04, Appendix A). Nonetheless, we evidenced the presence of Jun in the three fractions examined. Thus, densitometry showed that the subcellular location explained more than 16% of the total variance (*p* < 0.03, Figure 2 and Appendix A), with no significant influence of the sex or TDP-43 overexpression.

Finally, the analyses of the subcellular fractionation of murine brains revealed that the crude mitochondrial fractions, in line with the immunostaining measurements, contained a non-negligible amount of REST (Figure 3A). Crude mitochondria were the subcellular fraction with a higher concentration of REST. Regarding the influence of specific factors, neither sex nor TPD-43 overexpression was a significant factor contributing to the variance, in contrast with the subcellular location (39% of the variation, *p* < 0.0043, Appendix A). Besides a nonsignificant trend for an increase in REST in transgenic mice, we evaluated the potential effect of a mutated TARDBP overexpression in well-established REST targets by RT-qPCR. The results showed that one downstream REST, CYCS mRNA, was increased almost significantly in the TARDBP transgenic mice (Figure 3B).

To further confirm the interaction between TDP-43 and REST, we set up an in vitro model of TDP-43 and p-TDP-43 aggregation in human neural cells (N2A and SHSY-5Y). Thus, after sorbitol incubation (Figure 3C), both the number of TDP-43 aggregates and global intensity of TDP-43, but not p-TDP-43, were increased, with a high amount of TDP-43 in nuclear location. Thus, both the subcellular location (18% of the variance, *p* < 0.0001) and sorbitol incubation (11.2% of the variance, *p* < 0.0165) strongly influenced the TDP-43 immunostaining intensity. In this particular model, we explored the effect of REST targets on SHSY-5Y cells. The results of the RT-qPCR revealed that TDP-43 aggregation induced by sorbitol incubation was associated with significant decreases in the mRNA levels of SCN3B, FADD, PUMA, DAXX, SOD1, CAT, GAP43, 1433, MAPK11, MAPK12, and ARC, known REST targets. Sorbitol incubation also increased some REST targets such as NRX3 and ATP2B2 (Figure 3D).

To evaluate if TDP-43 aggregation can be related to mitochondrial dysfunction, we evaluated if sorbitol incubation in these conditions led to alterations in the cellular ATP production. The results (Appendix A) suggest that both mitochondrial and glycolysis-linked ATP production is severely affected by sorbitol incubation.

We also examined if other ALS-related cell stressors, such as ER stress, could induce similar delocalizations of the evaluated proteins. We have previously shown that proteasomal and ER stress induce a cytosolic mislocalization of TDP-43 [7]. Using similar conditions, we first evaluated if the cells exposed to proteasomal stress (epoxomicin) in similar conditions to the ones already reported to mislocalize TDP-43 also changed p-ERK and p-Jun. The results (Figure 4A) indicated that these proteins exhibited differential dynamics. Thus, in p-ERK, the nuclear levels were always inferior to the cytosolic ones (90% of the total variance explained by the cellular location, *p* < 0001).

Further, proteasomal stress induced by epoxomicin decreased the levels of p-ERK significantly (either at the nuclei and in the cytosol, 7% of the total variance, *p* < 0.0001). In p-Jun, there was a significant interaction between stress and the subcellular location, i.e., the effect on epoxomicin depended on the location. Therefore, epoxomicin treatment decreased the cytosolic levels of p-Jun (Figure 4A) in a close relationship with the increased levels in the nuclei (51% of the total variance explained by the interaction of stress and the subcellular location, *p* < 0.0001, Appendix A). In the case of ER stress (thapsigargin), we observed similar results in p-ERK. Thus, the total levels were decreased after the stress, both at the nuclear and cytosolic levels (Figure 4B). Similarly, the cytosol vs. nuclear location was the factor explaining the most variance (79% of the total variance, *p* < 0.0001, Appendix A). For p-Jun, while its preferential nuclear location was maintained, the ER stress induced by thapsigargin induced a significant early increase in the nuclei (similar to proteasome stress), but later on, the levels were decreased, in line with the changes in the cytosol (Figure 4B). Therefore, cytosol vs. nuclear location explained most of the variance (90% of variance, *p* < 0.0001, Appendix A).

We then evaluated the potential colocalization of these factors with the mitochondrial components. The results of the confocal microscopy (Figure 5 and Appendix A) suggested that the degree of colocalization was affected by cell stressors. In the case of proteasome stress, the degree of p-ERK colocalization increased significantly at the longer times evaluated (Figure 5A), while this was not present for p-Jun. Later, proteasome stress led to a decrease in the degree of colocalization (Figure 5A). For ER stress, at the shorter term, the z’-values increased for p-ERK, but later on, they showed a significant decrease (Figure 5B). Both at short and longer times, in the case of p-Jun, decreased degrees of colocalization were evident (Figure 5B).

To evaluate if the stress mentioned above could be behind the changes in the transgenic TDP-43 model, we also studied the effects of the proteasome and ER stress in the levels and distribution of REST in the non-neuronal cell line studied. For epoxomicin treatment, an early (2-h) increase in the REST intensity in the cytosol was followed by a decrease in the nuclear amount (at 4 h, Figure 6A). The degree of colocalization with the mitochondrial markers diminished as well (data not shown). Additionally, the results showed that ER stress increased the cytosolic and nuclear amounts of REST at longer times studied (Figure 6B) and also due to changes in the degree of colocalization with the mitochondria (z-values decreased by 20% in the epoxomicin treatment, data not shown).

## 3. Discussion

This work shows that several proteins implicated in ALS (TDP-43, ERK) and other neurodegenerative processes, such as AD (Jun, REST), show a sensitivity to cell stress. Several neurodegenerative diseases exhibit protein aggregates where these proteins may be present [19]. We evaluated three different major cellular stressors, namely oxidative stress, proteasome inhibition, and ER stress, and we demonstrated that the effects on cellular distribution showed stress and protein specificity.

In the case of TDP-43, the results showed that, after oxidative stress or osmotic stress, the cytosolic levels of this protein increased both in the endothelial cell line and in the neuronal cell lines. Further, the degree of colocalization within the mitochondria also increased. Of note, TDP-43 may show differences in behavior with p-TDP-43. Thus, while TPD-43 is accumulated after sorbitol stress in the cytosol, it was nonsignificant in p-TDP-43. The in vivo evidence presented shows that this is also the case in the brain lysates, where p-TDP-43 and TDP-43 show differential responses to the overexpression of mutated TARDBP. Whatever the case, the subcellular fractionation data agree with the enrichment of both endogenous murine tdp-43 and hTDP-43 in mitochondrially enriched fractions. Both the functional and morphological analyses showed that the conditions linked to p-TDP-43 aggregation (specially in sorbitol incubation) were associated with major changes in the mitochondrial ATP production. We should indicate that sorbitol incubation is quite harsh, which could impair the mitochondrial function by several pathways.

Regarding ERK, in response to increased oxidative stress, we noticed a decrease in the cytosolic and nuclear locations after a transient increase. Nonetheless, the degree of colocalization increased significantly, suggesting the close occurrence of p-ERK with mitochondrial epitopes again. In contrast with p-TDP-43, the slopes relating to nuclear and extranuclear p-ERK were inversely related to the oxidative stress intensity, suggesting the nuclear retention of this factor or rapid cytosolic clearance of it. The same phenomena (decreased both in the cytosol and in nuclei) was present after proteasome and ER stress. This later cell stress also decreased the colocalization within the mitochondrial epitopes. Interestingly, after the epoxomicin treatment, we observed an increase in the degree of mitochondrial colocalization. The subcellular fractionation in the TDP-43 model suggested that p-ERK could be significantly enriched with mitochondria, independently of the TARDBP overexpression.

In the present work we show, concerning p-Jun, no clear accumulation in the extranuclear location was present after oxidative stress, though the mitochondrial colocalization increased. After proteasome inhibition and ER stress, the extranuclear levels of this factor were decreased with concomitant retention in the nuclei (thought at later stages, the ER stress decreased the levels of nuclear p-Jun slightly). In clear contrast with oxidative stress, both ER stress and proteasome inhibition decreased the mitochondrial colocalization. Indeed, the subcellular fractionation suggested that, while Jun accumulated with crude mitochondria, this was not the case with p-Jun.

Regarding REST, its accumulation has been previously reported in a cytosolic location in neurodegenerative processes. Our data regarding the response to cell stress showed that its behavior was similar to p-Jun, showing an initial decrease in the cytosol followed by a slight increase in both the nuclei and cytosol. Epoxomicin treatment decreased the nuclear levels after the long exposure, while ER stress increased their values in the cytosol and nuclei. Indeed, the degree of colocalization increased after oxidative stress. Previous data also reported the relationship between changes in the REST expression and protein aggregates [20]. Particularly, several genes under the control of this negative regulator are upregulated, suggesting its impairment in human pathology [20]. Amongst the controlled genes, the authors indicated increased ubiquitin carboxy-terminal hydrolase L1, a component of the aggregates [20]. This protein is involved in the ubiquitin-proteasome pathway of proteostasis, suggesting that REST could influence proteostasis. Our data indicate that the reverse is also true, i.e., proteasome could control the REST protein levels and their subcellular location.

Interestingly, the in vivo data showed that the overexpression of mutated TARDBP increased the degree of REST enrichment in mitochondrial fractions. The interaction between TDP-43 alterations and REST is new, and it is reinforced by the fact that several REST-regulated genes appear affected under osmotic stress conditions, where TDP-43 is mislocalized. These genes included SCN3B, encoding a voltage-gated sodium channel [21], and two genes implicated in apoptosis (PUMA and FADD) [22], with downregulation in both cases. Interestingly, in the human cells evaluated, sorbitol decreased the expression of transcripts encoding antioxidant enzymes, such as SOD1 and catalase. Furthermore, in these cells, an increased expression of the ATP2B2 mRNA was found, encoding a plasma membrane Ca^++^ pump. Additionally, we noticed a decreased ARC mRNA expression in the cells, implicating altered mRNA traffic (one of the functions of TDP-43). We also detected an increased NRXN3 expression, encoding neurexin 3. This factor encodes a cell adhesion component whose paralogs have been recently implicated in synaptic strength, in close collaboration with the factors regulating RNA/aggregation toxicity [23]. We also detected a tendency for decreased p35 mRNA after osmotic stress, not achieving a signification threshold. Of note, P35 is a neuron-specific cyclin-dependent kinase 5 activator. When p35 is cleaved by calpain into p25, the protein is relocalized from the cell periphery to the nucleus and perinuclear region. Patients with AD accumulate the p25 form in their brain neurons [24]. This buildup is linked to an increase in CDK5 kinase activity, leading to abnormally phosphorylated microtubule forms. CDK5 kinase overactivation is linked to TDP-43 pathological effects in neuronal cells [25]. Noteworthy, we used proliferative cell lines (HMEC human mammary epithelial cells, N2 mouse neuroblastoma cells, and SHSY-5Y human neuroblastoma cells). It is known that these cellular types show a different response against oxidative and osmotic stress, besides having proliferation, which is not the case of neurons. Therefore, we need to show caution in the potential extrapolation of these results.

We also detected a tendency for increased *Foxo* 1 mRNA. It is known that this key metabolic transcription factor promotes neuron death [26]. Therefore, decreased values of antiapoptotic factors (PUMA and FADD) may be responses to this increased expression.

Accounting for the fact that one of their primary functions is working as transcription factors (in the case of ERK, Jun, and REST) or interacting with RNA (in the case of TDP-43), their localization in the extranuclear placements (particularly in close relationships with mitochondrial epitopes) suggests the existence of pathogenic mechanisms operating in common. All these factors require nucleocytosolic transport; therefore, a loss in this cellular property’s homeostasis may partially explain their presence in the cytosol. Indeed, we have recently reported that the nuclear envelope (where the proteins responsible for nucleocytoplasmic shuttling reside) showed altered properties in ALS patients and models [14].

Several reports have described the occurrence of TDP-43 in mitochondrial fractions [27,28,29] and other membranal fractions [18]. Regarding Jun, it is known that its N-terminal kinase (JNK) is located in the mitochondria [30]. Other studies demonstrate that the cytosolic location of c-Jun depends on its interaction with other transcription factors [31]. Of note, the major location of p-Jun in brain lysates is the nuclei, but we detected a consistent signal in the cytosol and in the crude mitochondrial fractions. Previous in vitro studies showed that c-Jun interacts with phospholipids [32]. Interestingly, other in vivo reports indicated that cytoplasmic c-Jun is associated with the mitochondria [33]. Indeed, it is known that c-Jun may interact with mitochondrial DNA motifs, the mitochondrial location validated by electron microscopy [34]. Despite some doubts for the mitochondrial location of typically nuclear transcription factors, such as NF-kB [35], it may be adequate to validate the reported findings further and establish the interaction of c-Jun with mitochondria. Further c-Jun binding sites occur within mtDNA genes and are negatively selected [34]. It has been suggested that oxidative phosphorylation, mitochondrial translation, and mtDNA repair processes.

Regarding p-ERK, it is known that there are mitochondrial substrates for its kinase activity [36]. Recent data underlines that ERK signaling specificity requires the spatial compartmentalization of ERK activity for signals like EGF to govern diverse functional responses via compartmentalized ERK activity [37]. Previous data showed that p-ERK mitochondrial location could be a consequence of tumoral transformation [38], demonstrating that a fraction of active ERK1/2 associates with succinate dehydrogenase and some mitochondrial chaperones, such as TRAP1 [39]. We have previously shown that motor neurons in the spinal cord from ALS patients exhibit p-ERK aggregates in extranuclear locations [7]. These aggregates may be related potentially to mitochondrial interactions. In this regard, we show that there are no interactions between p-ERK and TDP-43 overexpression in the evaluated murine model.

To the best of our knowledge, we have not found previous evidence of REST in the mitochondria. Previous evidence in human substantia nigra [10] showed a cytoplasmic staining profile in neurons. Nonetheless, in this publication, REST was found as aggregates closely related to autophagy impairment [40]. Thus, defects in the protein quality control system induce REST mRNA expression; its gene product mainly appears in aggregates. In brain subcellular fractionation experiments, we did not evidence changes in the total levels of REST. Nonetheless, we show that all stress evaluated, including ER, proteasomal, and oxidative stress, changes its cellular distribution, increasing the colocalization with mitochondrial epitopes. Indeed, recent data showed that CRISPR-mediated REST KO induced mitochondrial dysfunction and impaired mitophagy in vitro. Furthermore, REST overexpression impedes mitochondrial toxicity and mitochondrial morphology disruption through the transcription factor PGC-1α [41].

As for the limitations of our work, we must remark that this evidence was present in an endothelial-like cell phenotype. Therefore, neuronal or glial cells could exhibit different dynamics. Nonetheless, some of the findings in the endothelial cell culture were replicated independently in lysates of the brain cortex of a murine model, and in N2A and SHSY-5Y cells, human neuronal lines further validated the potential findings. We shall also indicate that the subcellular fractionation was more enriching than a strict purification of the indicated compartment. However, we can exclude nuclear contamination of the mitochondrial fraction (based on purity markers).

## 4. Conclusions

Cell stress related with ALS enhances the non-nuclear localization of transcription factors. These changes could be modeled in vitro with several cell types and may be related to impairments in nucleocytosolic traffic. These transcription factors, under cell stress, increase their interactions with the mitochondria and potentially influence their physiology (as shown for the TDP-43-REST interaction). In particular, TDP-43 aggregation was associated with alterations in the mRNA levels of the REST target genes involved in calcium homeostasis, apoptosis, and metabolism.

## 5. Materials and Methods

### 5.1. Animals

According to local laws and the Directive 2010/63/EU of the European Parliament, all experimental procedures were approved by the Institutional Animal Care Committee of the University of Lleida. The minimal number of animals was calculated according to the deviation of western-blot profiles in previous experiments [18]. Both non-transgenic and transgenic mice were obtained from JAX (The Jackson Laboratory, Bar Harbor, ME, USA). Transgenic mice were from the line B6.Cg-Tg (Prnp-TARDBP*Q331K)103Dwc/J (Stock number #017933). These mice express, employing the murine prion-promoter previously reported to drive transgene expression most abundantly in the central nervous system, both in neurons and astrocytes, ALS-linked mutant TDP-43 broadly throughout the central nervous system [42]. Under the control of the murine prion promoter, transgenic mice expressed the ALS-linked mutant of TDP-43 [43] (Q331K (glutamine to lysine substitution at amino acid position 331) fused to an N-terminal myc-tag. Animals employed here were from both sexes (at least *n* = 5 different mice from each sex) and from 90 days old, an age where the motor phenotype is not present. Housing and obtention of animals were as described [42,44]. For animal sacrifice, mice were anesthetized with 2.5% isoflurane. Brains were rapidly excised and maintained at 4 °C in isolation buffer for a maximum time of 15 min, being submitted to subcellular fractionation.

### 5.2. Subcellular Fractionation

Purification of nuclear, mitochondria and cytosol enriched fractions performed as described [45]. Briefly, brain samples were homogenized gently in isolation buffer (225-mmol/L mannitol, 25-mmol/L HEPES-KOH, and 1-mmol/L EGTA, pH 7.4 containing protease inhibitors (Cat #78429, Thermo Fisher Scientific, Waltham, MA USA) and Sodium Fluoride and Sodium Orthovanadate as phosphatase inhibitors) with 10–12 strokes in a glass tissue grinder. The homogenate was centrifuged for 10 min at 1500× *g* to obtain a nuclei-enriched fraction. The supernatant was washed twice by centrifuging for 10 min at 1500× *g* to eliminate any residual whole cells and cell debris. Supernatant obtained was centrifuged for 15 min at 10,000× *g*; the supernatant contained the ER/cytosol fraction, and the pellet contained the crude mitochondrial fraction. The ER/cytosol fraction was centrifuged for 1 h at 100,000× *g* in an ultracentrifuge and the supernatant was considered cytosol.

### 5.3. Cell Culture

Human mammary epithelial cells (HMEC) cell line (ATCC# PCS-600-010™, ATCC Manassas, VA, USA) was grown in DMEM medium (Invitrogen, Waltham, MA, USA,) supplemented with 10% fetal bovine serum heat inactivated (Invitrogen), 2-mM L-Glutamine (Invitrogen) and 20-U/mL penicillin and 20-µg/mL streptomycin (Invitrogen) as antibiotics. The cells were kept at 37 °C in humidify atmosphere with 5% of CO_2_.

N2A and SHSY-5Y (ATCC) cell lines were grown in Advanced MEM medium (Invitrogen) supplemented with 10% fetal bovine serum heat inactivated (Invitrogen), 2-mM L-Glutamine (Invitrogen) and 20-U/mL penicillin and 20-µg/mL streptomycin (Invitrogen) as antibiotics. The cells were kept at 37 °C in humidify atmosphere with 5% of CO_2_.

To study the effects of oxidative stress, alteration of endoplasmic reticulum, inhibition of proteasome activity and osmotic stress, cells were treated with 10-µM H_2_O_2_ (Millipore-Sigma, Burlington, MA, USA), 5-µM thapsigargin (Thp) (Millipore-Sigma), and 2.5-µM epoxomicin (Epox) (Millipore-Sigma), respectively, for 2 or 4 h or sorbitol (Millipore-Sigma) 0.4 M for 3 h. To avoid the influence of growth factors present in the fetal bovine serum on the results, the normal culture medium was replaced by Opti-Mem^TM^ (#31985062 Invitrogen) before 12 h before all assays, as described [7]. Cell viability was evaluated by employing the Prestoblue Cell Viability HS reagent (Thermo #P50200).

### 5.4. Indirect Immunofluorescence Analysis

Cells, seeded out on coverslips and incubated in serum and phenol red free medium (Opti-Mem^TM^, Invitrogen, Waltham, MA, USA) for 12 h were treated as indicated above. After incubation, cells were washed with PBS and then fixed with 3.7% paraformaldehyde for 10 min at room temperature. Cells were rinsed with PBS, permeabilized with 0.1% Triton X-100 in PBS for 30 min and subsequently blocked with 5% normal goat serum at room temperature for 1 h. Cells were incubated with (1) the mouse anti-p-TDP-43 (pS409/410) monoclonal antibody (TIP-PTD-M01) (diluted 1:200, Cosmo Bio Co., Tokyo, Japan); (3) the rabbit anti-p-ERK 1/2 polyclonal antibody (4370) (diluted 1:100, Cell Signaling, Beverly, MA, USA); the rabbit anti-REST polyclonal antibody (ab21635) (diluted 1:100, Abcam, Cambridge, UK); (4) the rabbit anti p-Jun polyclonal antibody (diluted 1:100, Cell Signaling, Beverly, MA, USA); and (5) the anti-ATP5A mouse monoclonal antibody (ab14748) (diluted 1:100, Abcam, Cambridge, UK) at 4 °C overnight. After 3 washes with 0.1% Triton X-100-PBS at RT for 10 min, cells were incubated with Alexa Fluor-488 goat anti-rabbit IgG or Alexa-Fluor-594 goat anti-mouse IgG (1:800, Molecular Probes, Eugene, OR, USA)-conjugated secondary antibody. Nuclei were stained with 4′,6′-diamino-2-phenylindole (DAPI) (1 mg/mL, Sigma, St. Louis, MO, USA). The coverslips were mounted in Fluoromount-G (Southern Biotech) and images were taken with an Olympus FV10i laser scanning confocal microscope and an Olympus FV1000 confocal microscopy, 60× magnification. For the evaluation of nuclear and cytosolic protein intensity, we used a dedicated pipeline created on the open source CellProfiler software [46]. Another CellProfiler pipeline was built to evaluate colocalization with the mitochondrial marker ATP5A. In both cases, 10 fields per condition were analyzed. CellProfiler pipelines employed are available in Appendix A. Mitochondrial network analyses were performed by employing the MINE software [47].

### 5.5. Western-Blot

Protein homogenates were prepared in the presence of inhibitors of phosphatases (Sigma-Aldrich) and proteases (Roche Applied Science, Penzberg, Germany). Protein concentration of the samples was measured using the Quick Start™ Bradford 1× Dye Reagent (Bio-Rad #5000205). Proteins were detected by immunoblotting using horseradish peroxidase-conjugated secondary antibodies and chemiluminescence (Santa Cruz Biotechnology). Protein samples were run on SurePAGE^TM^ Precast gels (4–20%, 15 wells GenScript, Piscataway, NJ, USA). Gels were blotted onto PVDF membranes by transfer at a constant 100 volts, 1 h at RT. The membranes were then blocked by nonfat dry milk solution (5%) in 1 × TBS (Tris Buffered Saline) and incubated in the desired primary antibody overnight.

Membranes were then washed in 1 × TBST with Tween 20 (0.05%) three times, 5 min each before incubating with the secondary antibody for 1 h at room temperature. Following the secondary antibody incubation, membranes were washed in 1 × TBST with Tween 20 (0.05%) three times, 5 min each and one time, 5 min, with TBS 1 ×. Blots were imaged in the Chemidoc MP Imaging System following incubation in the Immobilon ECL Ultra Western HRP Substrate (Merck Millipore, Burlington, NJ, USA).

For Jun/ p-c-Jun Westerns, protein samples were run on SurePAGE^TM^ Precast gels (4–20%, 15 wells GenScript, Piscataway, NJ, USA). Cruz Marker™ molecular weight standards (sc-2035, Santa Cruz Biotechnologies, Dallas, TX, USA) were loaded in the gels. Gels were blotted onto low fluorescence PVDF membranes (Immobilon^®^-FL PVDF: sc-516541). Nonspecific binding was blocked in incubating membranes with UltraCruz^®^ Blocking Reagent (sc-516214) for 1 h at room temperature, with shaking.

The blocked membranes were incubated with the appropriates Alexa Fluor^®^ conjugated primary antibodies (Anti-p-c-Jun Antibody (KM-1) Alexa Fluor^®^ 790 and Anti c-Jun Antibody (G-4) Alexa Fluor^®^ 680, from Santa Cruz) diluted 1:1000 in UltraCruz^®^ Blocking Reagent. Cruz Marker™ MW Tag-Alexa Fluor^®^ 680 (sc-516730) and Cruz Marker™ MW Tag-Alexa Fluor^®^ 790 (sc-516731) at 1:1000 were added to obtain molecular weight distribution.

Membranes were incubated in this mixture for 2 h at room temperature, in the dark, with shaking. Membranes were then washed three times for 5 min each with TBST and once for 5 min with TBS. Blots when then placed on the top of blotter paper and dried for 5–10 min. The western blot was imaged using the infrared (IR) laser-based instrumentation LI-COR Odyssey (Lincoln, NE, USA).

### 5.6. RT-QPCR

RNA was extracted from cells and brain lysates using TRI Reagent (Thermo Fisher Scientific, Waltham, MA, USA, AM9738) following the manufacturer’s instructions. RNA concentrations were measured using a NanoDrop ND-1000 (Thermo Fisher Scientific). One microgram of RNA was used for retrotranscription to cDNA employing TaqMan Reverse Transcription Reagent and random hexamers (Thermo Fisher Scientific, N8080234).

RT-qPCR experiments were performed using a CFX96 instrument (Bio-Rad, Hercules, CA, USA) with SYBR Select Master mix for CFX (Thermo Fisher Scientific, #4472937). Each 20 µL reaction mix contained 4-µL cDNA, 10-µL SYBR Select Master Mix, 0.2 nM of forward primer and 0.2 nM of reverse primer solutions and 4-µL PCR grade water. RT-qPCR run protocol was as follows: 50 °C for 2 min and 95 °C for 2 min, with the 95 °C for 15 s and 60 °C for 1 min steps repeated for 40 cycles, and a melting curve test from 65°C to 95 °C at a 0.1 °C/s measuring rate. Primers employed in these experiments, previously described in reference [9] are listed in Appendix A.

### 5.7. Mitochondrial Function Analyses

Mitochondrial function was estimated employing the Seahorse XF HS Mini Analyzer (Agilent technologies). Briefly, to obtain an effect of the aggregation-prone conditions in the mitochondrial function we employed the Seahorse XFp Real-Time ATP Rate assay kit (Agilent#103591-100), according to the manufacturer’s instructions. 20,000 SHSY-5Y cells were plated in the microplates of the kit and incubated with sorbitol (3 h, 0.4 M). After this incubation, oxygen consumption and extracellular acidification, were recorded, in basal conditions, and after oligomycin and rotenone/antimycin addition.

### 5.8. Statistical Analysis

All statistics were performed using the GraphPad Prism version9.1.2 for Windows software (GraphPad Software, San Diego, CA, USA). Differences between groups were analyzed by the Student’s *t*-tests, One-way, Two-way, and Three-way ANOVA analyses, with adequate post-hoc analyses, once normality of variables was tested by Kolmogorov–Smirnov test. The 0.05 level was selected as the point of minimal statistical significance in every comparison.

## Figures and Tables

**Figure 1 ijms-22-08853-f001:**
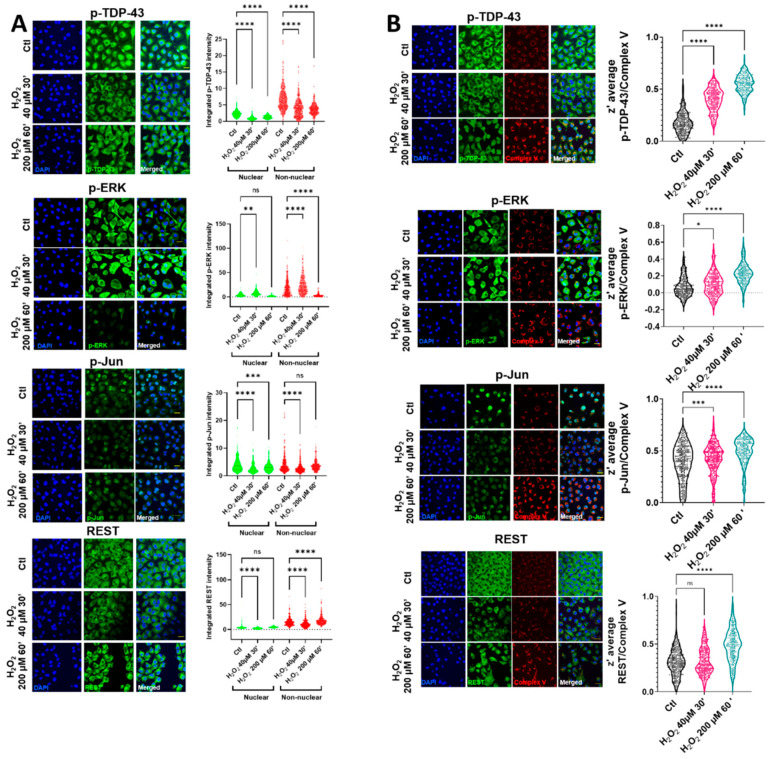
Oxidative stress induces changes in the levels of proteins implicated in neurodegeneration and its colocalization with mitochondrial epitopes. (**A**) Representative confocal microscopy images of HMEC cells immunostained with antibodies against p-TDP-43 (left), p-ERK (middle left), p-Jun (middle right), and REST (right), showing diverse effects of oxidative stress (H_2_O_2_, dose and time indicated) in nuclear and non-nuclear (cytosol) immunostaining (quantified below). (**B**) Representative coimmunostaining confocal microscopy images of the above-mentioned proteins, with mitochondrial epitopes (Complex V). The degree of colocalization was estimated by calculation of the z’ factor, shown in the violin graphs below. In (**A**), the bars indicate the mean with the standard deviation shown by the lines (*n* = 200–296 cells for p-TDP-43, *n* = 191–255 for p-ERK, *n* = 234–326 for p-Jun, and *n* = 217–415 for REST, obtained in at least 4 independent replicates). * Indicates *p* < 0.05, ** *p* < 0.01, *** *p* < 0.001, and **** *p* < 0.0001 by Sidak’s post-hoc multiple comparison test after a 2-way ANOVA (in **A**) or by Dunnett’s post-hoc multiple comparison test after an ANOVA (in **B**). Bars in (**A**,**B**) are 60 micrometer long.

**Figure 2 ijms-22-08853-f002:**
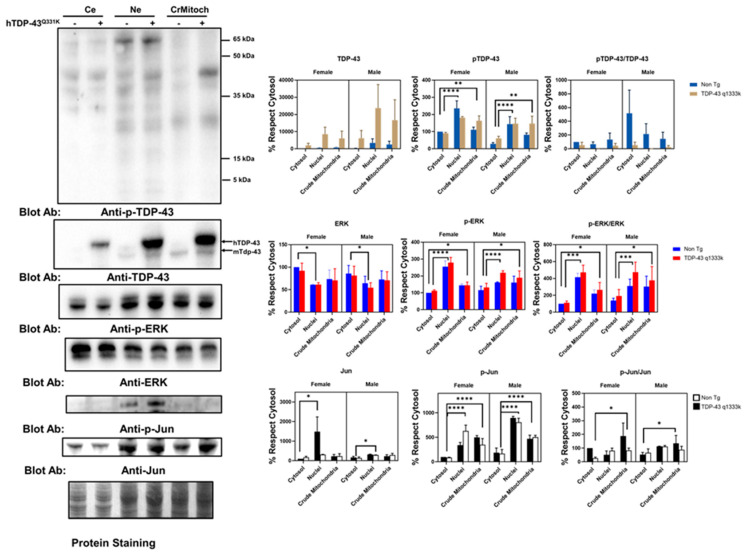
Cellular subfractionation evidence for the in vivo colocalization of the proteins implicated in neurodegeneration with the mitochondrial components. As shown by the Western blot analyses of brain lysates after subcellular fractionation, in addition to the nuclear-enriched (Ne) and cytosolic-enriched (Ce) compartments’ crude mitochondrial fractions (CrMitoch), both non-transgenic and transgenic hTDP-43 mice show the presence of p-TDP-43, p-ERK, and Jun. The levels were quantified by densitometry in the brains from 90-day-old mice. The Western blots shown are for female specimens. Right panels indicate the quantitative analyses. Bars indicate the mean values with lines showing the standard deviation. * Indicates *p* < 0.05, ** *p* < 0.01, *** *p* < 0.001, and **** *p* < 0.0001 by Dunnett’s post-hoc multiple comparison test after a three-way ANOVA (*n* = 4 different mice from each genotype and sex). The ANOVA values are shown in the text.

**Figure 3 ijms-22-08853-f003:**
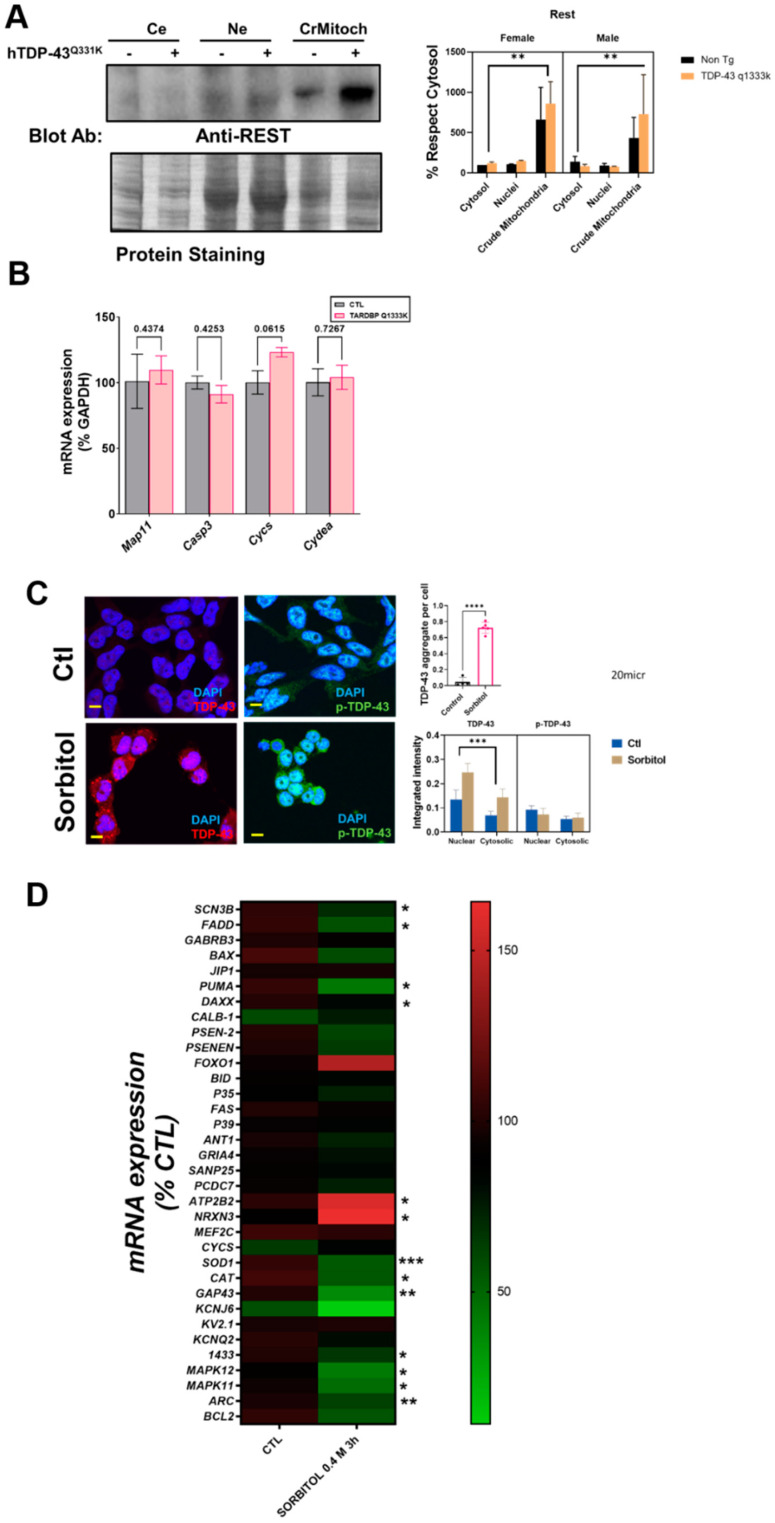
Cell stress-induced mislocalization of TDP-43 is associated with changes in REST-regulated genes. (**A**) Shows representative Western blots from brain lysates, indicating that REST is localized within crude mitochondria (CrMitoch), in comparison with enriched nuclei (Ne) or cytosolic extract (Ce). Levels were quantified by densitometry in brains from 90-day-old mice (right panel). Western blots shown are for female specimens. Bars indicate the mean values with lines showing the standard deviation. (*n* = 4 different mice from each genotype and sex). (**B**) Shows the effect of mutated TARDBP overexpression in the mRNA levels of REST-regulated genes in brain lysates, quantified by RT-qPCR. Bars indicate mean values with lines showing the standard deviation (*n* = 3 mice from each genotype). (**C**) Shows representative immunofluorescence images of N2A cells under osmotic stress (Sorbitol, 0.4 M, 4 h), demonstrating that both TDP-43 and p-TDP-43 are localized in a non-nuclear location as aggregates after sorbitol incubation (graphs in the right panel). (**D**) Shows a heatmap of the mRNA expression levels of REST-regulated genes in SHSY-5Y cells under osmotic stress, with the scale on the right showing the relative overexpression (in red) or downregulation (in green), quantified by RT-qPCR. * Indicates *p* < 0.05, ** *p* < 0.01, *** *p* < 0.001, and **** *p* < 0.0001 by Dunnett’s post-hoc multiple comparison test after a three-way ANOVA (in (**A**) for the integrated intensity of p-TDP-43 or TDP-43 in confocal immunofluorescence analyses or densitometry in (**C**)), for the Student’s *t*-test (in (**C)** for the number of aggregates; *n* = 150 different cells quantified in each condition or 4 independent RT-qPCR experiments in (**B**) or in (**D**)). The bars in (**C**) are 20 micrometer long.

**Figure 4 ijms-22-08853-f004:**
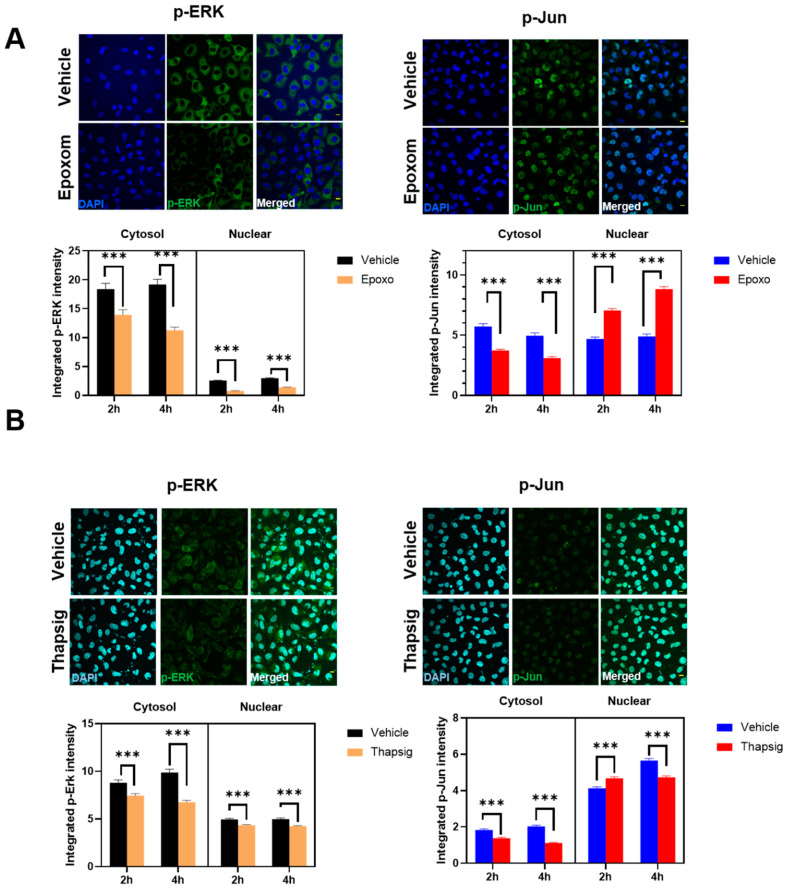
Proteasome and ER stress induces changes in the levels of p-ERK and p-Jun. Representative confocal microscopy images of HMEC cells immunostained with antibodies against p-ERK and p-Jun showing diverse effects of proteasome inhibition (epoxomicin) in (**A)** and ER stress (thapsigargin) in (**B**) in nuclear and non-nuclear (cytosol) immunostaining (quantified below). Images shown are for 2 h of incubation. In (**A**), the bars indicate the mean with the standard deviation shown by lines (*n* = 195–283 cells for p-ERK and *n* = 285–394 for p-Jun) and, in (**B**), *n* = 519–658 cells for p-ERK and *n* = 415–553 for p-Jun. *** Indicates *p* < 0.001 by Sidak’s post-hoc multiple comparison test after a 2-way ANOVA. The bars in the (**A**,**B**) micrographs are 20 micrometer long.

**Figure 5 ijms-22-08853-f005:**
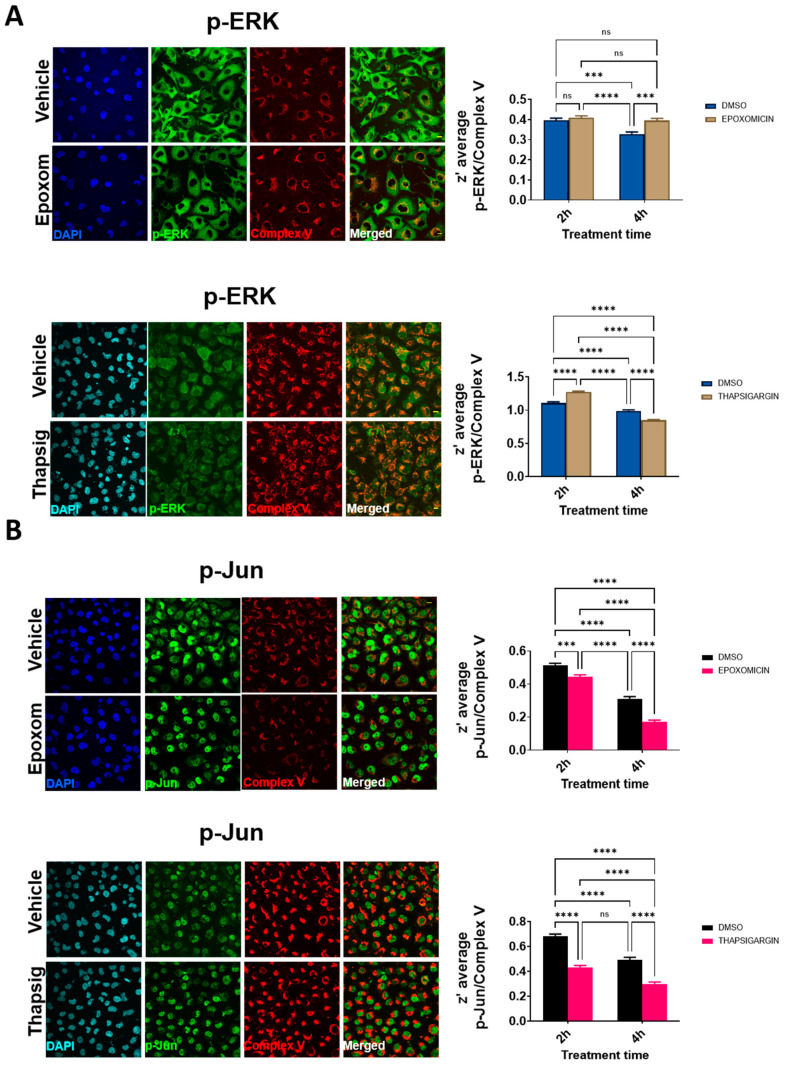
Proteasome and ER stress induces changes in the colocalization degree of p-ERK and p-Jun with mitochondrial epitopes. Representative confocal microscopy images of HMEC cells immunostained with antibodies against p-ERK and mitochondrial epitopes in (**A)** and against p-Jun and mitochondrial epitopes in (**B**), showing diverse effects of proteasome inhibition (epoxomicin) and ER stress (thapsigargin) in the degree of colocalization estimated by calculation of the z’ factor (right panels). Images shown are for 2 h of incubation. Bars indicate mean with standard deviation shown by lines (*n* = 195 to 658 cells for p-ERK and *n* = 285–553 for p-Jun); *** indicate *p* < 0.001, and **** *p* < 0.0001 by Bonferroni’s post-hoc multiple comparison test after 2 way ANOVA. Bars in (**A**,**B**) micrographs are 20 micrometer long.

**Figure 6 ijms-22-08853-f006:**
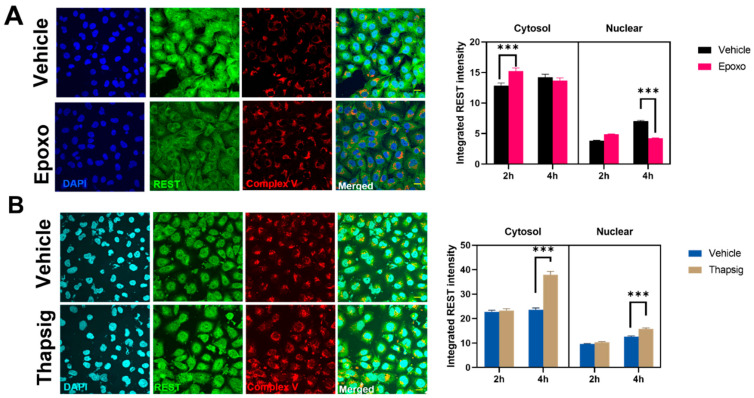
ER and proteasome stress induces changes in the levels of REST and its colocalization with the mitochondrial epitopes. Representative confocal microscopy images of HMEC cells immunostained with antibodies against REST, and the mitochondrial complex V showing the diverse effects of proteasome stress (**A**) and ER stress (**B**) (*n* = 365–559 different cells). Images shown are for 2 h of incubation. In the graphs on the right, the bars indicate the mean with the standard deviation shown by lines. *** Indicates *p* < 0.001 by Sidak’s post-hoc multiple comparison test after a 2-way ANOVA. The bars in the (**A**,**B**) micrographs are 40 micrometer long.

## Data Availability

Raw data are available from the authors on reasonable request.

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
