# Peer review of "Cell Stress Induces Mislocalization of Transcription Factors with Mitochondrial Enrichment"

_ijms, 2021, doi:10.3390/ijms22168853_

Round 1

Reviewer 1 Report

The aim of the study by Rossi C et al titled “Cell Stress Induces Mislocalization of Transcription Factors with Mitochondrial Enrichment“ was to examine the extranuclear inclusions of transcription factors (ERK,  Jun, TDP-43 and REST) in a cellular model of oxidative stress.

The hypothesis is based on previous studies, is well explained and experimental designs are rigorous. In the context of ALS the author have previously found that cellular stress induces TDP-43 pathological changes associated with ERK1/2 dysfunction.

Their main results have corroborated that the interaction of TDP-43, REST, Jun, and ERK with extranuclear components, such as mitochondria, is dependent on cell stress. Moreover, their novel results have demonstrated the link between TDP-43 pathology with changes in REST-dependent mRNAs.

Their final conclusion has been that cell stress increased mitochondrial translocation of nuclear proteins, increasing the chance of proteostasis alterations. Thus, the authors show that TDP-43 aggregation impacts REST target genes, disclosing an exciting interaction between these two transcription factors in neurodegenerative processes.

The Discussion is clear and full explained. However, I suggest that authors review the manuscript. My suggestion is that the authors should contribute new results or change these to strengthen the conclusion of their research.

Main comments

  1. In line 49 the authors mentioned that “In the case of ALS, TDP-43 is a primary component of these aggregates”. In Introduction Section the authors mentioned the cytoplasmic aggregation of these transcription factor appear in several neurodegenerative disorders (ALS, AD, PD). In lines 87-89 the authors mentioned that “we focused on the potential interaction with mitochondria due to the relevance of mitochondrial (dys)function in neurodegenerative conditions (particularly in ALS). They should discuss if it is a common mechanism in all neurodegenerative diseases or if the aggregation of a transcription factor is specific to a certain disease.

  1. In lines 92-93 the authors indicated that “we demonstrate an unreported association of TDP-43 pathology and changes in REST-dependent mRNAs”. They could explain the relationship between REST (a transcription factor identified in AD and PD) with their stress oxidative model and its relevance in ALS.

  1. In order to clarify the toxic effect of TDPs aggregation and loss mitochondrial function the authors should analyze cellular viability in their oxidative stress model (i.e MTT, Annexin, LDH).

  1. In lines 94-95 the authors mentioned that “the enhanced mitochondrial interaction of these proteins could contribute to the reported loss of mitochondrial functions”. Accordingly, the authors should use a method of mitochondrial function analysis (i.e mitochondria respiratory complex I or IV enzymatic activity) to reenforce the conclusion.

  1. The authors should add the scale bars in the images shown in all figures.

  1. In order to clarify the nuclear or cytosolic location of transcription factors, in Figure 4B the authors should include higher magnification from immunostaining images.

  1. In lines 329-234 the authors mentioned the p35 and cdk5 role in neurons cells (post-mitotic cells), but they have used proliferative cell lines (HMEC human mammary epithelial cells, N2 mouse neuroblastoma cells and SHSY-5Y human neuroblastoma cells). It is known that these cellular types show a different response against oxidative stress. The authors should discuss it.

Reviewer 2 Report

Dear Authors,

This is a very good work demonstrated how cell stress induced mitochondrial translocation of nuclear proteins. The experiments are appropriate for the goals of the study. I only have a few comments:

-Please, review the references on the discussion. I think there are some missing, like in line 305, and 347 (I suppose you refer to a previous work, but I have not seen the reference). 
- I know the conclusions are not mandatory, but I suggest at least include a paragraph on the discussion devoted only to conclusions. The last paragraph mixed the limitations of the study with a conclusion sentence.

Thank you very much. 

Author Response

Please see the atachment

Reviewer 3 Report

Mitochondrial dysfunction was observed in many neurodegenerative conditions. In this manuscript, the authors investigated the cellular distribution of some transcription factors in stressed epithelial cell line. Also, they investigated the potential interaction of the nuclear transcription factors with mitochondria in TDP-43 Q331K mice model under different cellular stress. This is a very interesting study that could enhance our understanding for mitochondrial dysfunction in ALS. However, most of the provided figs (images and graphs) were with low resolutions, even with zooming I could not read/see them clearly. For example, graphs in figs 1A and fig2 are blurry. Most of Confocal images were small, so I could not see the details clearly. The font size was not consistent in all graphs. For example, font sizes in graphs in Fig 1 were different.  The authors must provide high-resolution figs.

Some minor issues:

  1. All in vivo and in vitro should be written Italic
  2. In page 2, line 59, what does SGs stand for? The authors should define SGs
  3. Page 10, line 281, P-TDP-43, P should be lowercase
  4. In page 12, line 348, more references should be added as the authors state that “several reports..”
  5. In page 12, line 388, N2a, a should be uppercase and the same case for y in SHSY-5y

Round 2

Reviewer 1 Report

Thank you for the authors. In this new version, Rossi C et al have revised the Ms and they have made corrections and add new convincing data, which reinforces their conclusions.

Author Response

We thank the reviewer for this appreciation

Reviewer 3 Report

All my comments have been addressed 

Author Response

We thank the reviewer for the effort devoted in our manuscript